# Solubility Determination and Comprehensive Analysis of the New Heat-Resistant Energetic Material TNBP

**DOI:** 10.3390/molecules28062424

**Published:** 2023-03-07

**Authors:** Luoluo Wang, Minchang Wang, Ying Kang, Yong Zhu, Hai Chang, Ning Liu

**Affiliations:** 1Xi’an Modern Chemistry Research Institute, Xi’an 710065, China; 2State Key Laboratory of Fluorine & Nitrogen Chemicals, Xi’an 710065, China

**Keywords:** 4,8-bis(2,4,6-trinitrophenyl)difurazolo [3,4-b:3′,4′-e]pyrazine, solubility, dissolution behavior, thermodynamic properties

## Abstract

To improve the crystal quality of 4,8-bis(2,4,6-trinitrophenyl)difurazolo [3,4-b:3′,4′-e] pyrazine (TNBP), the solubility of TNBP in organic solvents (six pure and four mixed solvents) was determined by the laser monitoring technique from 293.15 to 353.15 K. The results showed that the solubility was positively correlated with the increase in the experimental temperature and the main solvent content, except for the co-solvent phenomenon in the DMSO + ethyl acetate solvent mixture. To explain the dissolution behavior of TNBP, the KAT-SER model was analyzed for pure solvent systems, and it was found that hydrogen bonding alkalinity and self-cohesiveness were the main influencing factors. The free energy of solvation and radial distribution function of TNBP in mixed solvents were also obtained by molecular dynamics simulation, and the effect of solute–solvent and solvent–solvent interactions on the solubility trend was analyzed. The experimental data were correlated using three empirical equations (van’t Hoff equation, modified Apelblat equation, and λh equation), and the deviation analysis showed the good applicability of the modified Apelblat model. Furthermore, the dissolution of TNBP was heat-absorbing and not spontaneous, according to the thermodynamic characteristics estimated by the van’t Hoff equation.

## 1. Introduction

4,8-Di(2,4,6-trinitrophenyl) difurazano [3,4-b:3′,4′-e]pyrazine (TNBP, Figure 1), first synthesized by Russian scientists in the 1990s [1], has become an ideal choice for ultra-high temperature heat-resistant energetic materials due to its excellent heat resistance and good detonation performances (T_d_: 415 °C, D: 7874 m/s, P: 28.2 GPa). Currently, the morphology of TNBP synthesized by the process has obvious angularity and uneven particle size distribution, affecting the product performances [2]. Therefore, improving the crystal quality of TNBP is of great importance for its further application in the military industry.

Solution recrystallization enables energetic materials to achieve the desired crystalline state and properties. However, the existing reports on TNBP mainly focus on optimizing the synthesis process [3], and there are few studies on recrystallization [4,5], especially the crystallization thermodynamics data for TNBP. Crystallization thermodynamics, as the basis of energetic material recrystallization research [6,7], can provide a basis for the selection of crystallization solvents and methods and has practical significance for the development and optimization of the TNBP crystallization process.

In this work, the molar fraction solubility of TNBP in six pure solvents (dimethyl sulfoxide (DMSO), N, N-dimethylformamide (DMF), N-methylpyrrolidone (NMP), acetone, acetonitrile, ethyl acetate) and the mixture of DMSO and four other solvents (water, acetone, acetonitrile, ethyl acetate) were determined. The experimental solubility was correlated using the van’t Hoff equation, modified Apelblat equation, and λh equation to broaden the applicability of the obtained data. Moreover, the factors affecting the solubility behavior of TNBP in pure solvents and mixed systems were analyzed using the KAT-LESR model and molecular dynamics simulations, respectively. In addition, the thermodynamic characteristics of TNBP throughout the dissolution process were also examined.

## 2. Results and Discussion

### 2.1. Solid-State Properties of TNBP

The PXRD patterns of crystals obtained before and after crystallization are given in Figure 2a. The crystals obtained by the crystallization of TNBP in all solvent systems have the same crystal phase compared with the raw material, which has characteristic peaks at 2θ of 10.4°, 12.9°, 14.6°, 15.1°, 16.3°, 18.2°, 18.6°, 18.9°, 20.8°, 21.9°, 22.3°, 23.1°, 23.8°, 24.7°, 25.1°, 25.8°, 26.4°, 29.6°, 30.2° and 20.6°.

According to the TG-DTG curve of TNBP (Figure 2b), it can be seen that the exothermic peak of TNBP is at 415.4 °C, and there is only a decomposition process during the heating process. To obtain its melting point (T_m_) and enthalpy of melting (Δ_fus_H), they are estimated by the additive and non-additive group contribution method (Appendix A) [8,9], and the results show that T_m_ is 392.23 °C and Δ_fus_H is 50.50 KJ·mol^−1^.

### 2.2. Pure Solvent Solubility Results

The experimental data on the molar fraction solubility of TNBP in six pure solvents are shown in Table 1 and plotted in Figure 3. TNBP’s solubility in pure solvents increases with rising temperature, indicating that the dissolution process is heat absorbing, and the temperature increase is conducive to the dissolution equilibrium. At lower temperatures (293.15–323.15 K), the solubility is in the following order from largest to smallest: NMP, DMF, DMSO, acetone, acetonitrile, and ethyl acetate. The change in solubility of TNBP in DMSO with temperature is relatively apparent. Therefore, when the temperature is higher than 333.15 K, the solubility of TNBP in DMSO is greater than that in DMF.

From Appendix A, the polarity order of the six solvents is: acetonitrile > DMSO > DMF > NMP > acetone > ethyl acetate [10], which does not correspond to the true order of solubility in all solvents. Therefore, polarity is not the main factor in determining the solubility of TNBP in pure solvents. To further study the solvent effect, the solubility of TNBP in pure solvents at 313.15 K is fitted using the KAT-LSER model (SPSS software 17.0). The α, β, π^⁎^, and δH^2^ values for the solvents are listed in Appendix A [11], and the density of TNBP (1.980 g/cm^3^) can be found in the SciFinder database. The results are shown in Equation (1):$$\mathrm{lnx}=-7.570\left(0.091\right)+3.930\left(0.275\right)\pi^{*}+10.936\left(0.406\right)\beta$$
$$+7.314\left(0.691\right)\alpha \;-\;11.761(0.385)\frac{V_{s}\delta_{H}^{2}}{100\mathrm{RT}}$$
(1)$$N\;=\;6\;R^{2}\;=\;0.998\;\mathrm{RSS}\;=\;0.001\;F\;=\;4162.13$$
where n represents the six pure solvents, R^2^ is the correlation coefficient, RSS represents the residual sum of squares, and the F-test is the ratio of variance test. The numbers in brackets refer to the standard deviation of the model coefficients. As can be seen from the above equations, the relative contributions of π^⁎^, β, α, and δH^2^ to the solubility of TNBP are 9.47%, 26.34%, 17.62%, and 28.33%, respectively. Among them, the parameter coefficients of π^⁎^, β, and α are all positive, indicating that the solubility of TNBP increases with increasing dipole/polarizability parameter, hydrogen bonding alkalinity, and hydrogen bonding acidity of the solvent. The δH^2^ parameter coefficient is negative, indicating that the self-cohesiveness of the solvent plays an opposite role in the dissolution process of TNBP.

In addition, the van’t Hoff equation, modified Apelblat equation, and λh equation all fit well with the experimental values of TNBP solubility in pure solvents. According to Appendix A, the mean 100RAD values for the three models are 1.92, 1.07, and 1.68. The mean 1000RMSD values are 0.30, 0.07, and 0.19, and the mean R^2^ values are 0.9546, 0.9907, and 0.9709, respectively. The correlation order for the three equations is modified Apelblat > λh > van’t Hoff. Consequently, the modified Apelblat equation correlates better (Figure 3) and is a guide for fitting and extending the application of experimental solubility data.

### 2.3. Mixed Solvent Solubility Results

As DMSO is an excellent solvent for solubilizing TNBP, studying the solid–liquid phase equilibrium of DMSO with the four non-solvents (water, acetone, acetonitrile, and ethyl acetate) can provide more options for the crystallization process of TNBP. The experimental values of the molar fraction solubility of TNBP in four mixtures of DMSO with water, acetone, acetonitrile, and ethyl acetate at different temperatures and mass fractions are shown in Table 2, Table 3, Table 4 and Table 5 and plotted in Figure 4. The solubility of TNBP in mixed solvents varies with solvent composition and temperature. For all the mixed solvents studied, solubility is positively correlated with increasing temperature. In the three systems (DMSO + water, DMSO + acetone, and DMSO + acetonitrile), the solubility of TNBP increases with the increasing mass fraction of DMSO when the temperature is constant. However, the co-solvent phenomenon occurs in the mixture of DMSO + ethyl acetate, which means that the solubility of TNBP reaches a maximum when the mass fraction of DMSO is 0.9, after which the solubility decreases with the addition of DMSO. The discovery of this phenomenon has great significance for controlling the crystallization operation range of TNBP in this system [12,13].

In order to better understand the reasons for the appearance of the co-solvent phenomenon, intermolecular interactions obtained using molecular dynamics simulations are explained. The solute–solvent intermolecular interactions can be quantitatively described by the calculated solvation free energies. Table 6 lists the solvation free energy of TNBP in four solvent mixtures, and its absolute value increases with the increase in DMSO mass fraction. The solvation free energy in the DMSO + ethyl acetate system likewise peaks when ω_DMSO_ is 0.90. Therefore, it can be concluded that the order of the magnitude of the absolute value of the solvation free energy follows the same trend as the solubility change of TNBP. It is further deduced that the stronger the solute–solvent interaction, the greater the solubility.

Competition between the solvent–solute and solvent–solvent interactions can be analyzed using the radial distribution function (RDF). RDF peaks between 1.5 and 3.5 Å are hydrogen bonding interactions, and peaks in the range 3.5–5.0 Å represent van der Waals forces. Appendix A shows the RDF curves between solute–solvent. Hydrogen bonds are formed between solute–solvent in all four systems with strong interactions. The solvent–solvent RDF curves (Figure 5) reflect that the intermolecular forces are also stronger in the DMSO + water, DMSO + acetone, and DMSO + acetonitrile systems. The peak intensity becomes stronger with increasing DMSO content. These indicate that both solute–solvent and solvent–solvent interactions strongly influence the solubility of TNBP in the three systems. However, in the DMSO + ethyl acetate mixture, the solvent–solvent intermolecular interaction is weaker, with the peak at 4.93 Å, and the peak intensity is essentially constant. The above results suggest that the solute–solvent interaction in this system is less affected by the solvent–solvent interaction, which leads to the co-solvent phenomenon.

In addition, the co-solvent phenomenon of the DMSO + ethyl acetate system was further investigated using infrared spectroscopy experiments. Figure 6a shows the comparative IR spectrum of TNBP dissolved in mixed solutions with different mass fractions of DMSO. As ω_DMSO_ increases, the carbonyl peak gradually moves toward the lower wave number, and the positions of the other peaks do not change. Considering that the nitro group’s oxygen atom in TNBP and ethyl acetate’s hydrogen atom form intermolecular hydrogen bonds, the electron absorption of the nitro group affects the carbonyl group in ethyl acetate, resulting in the apparent redshift of the peak. The above results prove the existence of hydrogen bonding between TNBP and the mixed solution. Figure 6b compares the IR spectra between the solvents and shows that all the carbonyl peaks are around 1742 cm^−1^ and do not shift, indicating little hydrogen bonding between the solvent–solvent. These are in good agreement with our calculation results.

The solubility data of TNBP in mixed solvents were correlated using the three thermodynamic equations. The results of the model parameters and the correlation data are shown in Appendix A. In the DMSO + water system, the maximum 100ARD value for the van’t Hoff, modified Apelblat, and λh equations are 20.18, 32.95, and 34.18, respectively. The 1000RMSD and R^2^ are not significantly different, which suggests that the van’t Hoff equation is more applicable to this system. In the other binary mixed systems, the mean values of 100ARD for the three models are 1.61, 0.50, and 1.35, respectively. The mean values of 1000RMSD are 0.65, 0.21, and 0.53, respectively, and the mean values of R^2^ are 0.9599, 0.9908 and 0.9729, respectively. Therefore, the modified Apelblat equation is more suitable for the DMSO + acetone, DMSO + acetonitrile, and DMSO + ethyl acetate systems. These can predict the solubility data of other temperature points in the system and provide a theoretical basis for future crystallization processes.

### 2.4. Thermodynamic Properties of Dissolution

The enthalpy of dissolution (Δ_dis_H), the entropy of dissolution (Δ_dis_S), and the Gibbs free energy (Δ_dis_G) of TNBP in each solvent system are shown in Appendix A. In pure solvents, Δ_dis_H is positive, indicating that the dissolution of TNBP is a heat-absorbing process and that the interaction between TNBP molecules and solvent molecules is stronger than between solvent molecules. Δ_dis_G is also positive, and the Gibbs free energy increases in the same order as the decrease in solubility (Figure 7a), implying that the dissolution process is non-spontaneous. In the solvent mixture, both Δ_dis_H and Δ_dis_G are also positive, and Δ_dis_G decreases with increasing DMSO content in the solvent mixture, which means that the dissolution of TNBP in the chosen solvent mixture is a “heat-absorbing and non-spontaneous” process.

## 3. Experimental

### 3.1. Materials and Instruments

The mass fraction of 0.980 for the yellow solid of TNBP was provided by the Xi’an Modern Chemistry Research Institute (Xi’an, China). All organic solvents, including DMSO, DMF, NMP, acetone, acetonitrile, and ethyl acetate, were analytical grade. Deionized water was homemade in the laboratory.

The D-MAX 2500 powder X-ray diffractometer (PXRD, Rigaku, Japan) was used to characterize the TNBP samples obtained by crystallization in different systems; the METTLER differential scanning calorimeter (Mettler Toledo, Switzerland) was used to measure the thermal properties of the TNBP recrystallized samples.

### 3.2. Solubility Determination

In this work, the dynamic laser monitoring method was used to measure the solubility of TNBP in six pure solvents (DMSO, DMF, NMP, acetone, acetonitrile, and ethyl acetate) and four mixtures of DMSO with acetone, acetonitrile, and ethyl acetate, respectively. The method is less time-consuming, quicker, and simpler than the traditional static methods, and there is no need to take samples for analysis. In addition, the accuracy of the test can meet the requirements.

The molar fraction solubility of TNBP in the pure solvent is shown in Equation (2); the molar fraction solubility of TNBP in mixed solvents is shown in Equation (3); the molar fraction of DMSO in mixed solvents is shown in Equation (4).
(2)$$x=\frac{m_{1}{/M}_{1}}{m_{1}{/M}_{1}{+m}_{2}{/M}_{2}}\;$$
(3)$$x=\frac{m_{1}{/M}_{1}}{m_{1}{/M}_{1}{+m}_{3}{/M}_{3}{+m}_{4}{/M}_{4}}\;$$
(4)$$x_{\mathrm{DMSO}}=\frac{m_{3}{/M}_{3}}{m_{3}{/M}_{3}{+m}_{4}{/M}_{4}}\;$$
where m_1_ and m_2_ are the masses of TNBP and solvent, g; M_1_ and M_2_ are the molar masses of TNBP and solvent; m_3_ and m_4_ are the masses of DMSO and non-solvent (acetone, acetonitrile, ethyl acetate), g; M_3_ and M_4_ are the molar masses of DMSO and non-solvent.

The experimental determination process is as follows: a certain mass of solvent and an excess of TNBP are accurately weighed using the analytical balance (ME204/02, Mettler Toledo, Switzerland) and placed in a reactor. The temperature is increased at the specific rate (1 K·h^−1^), and the transmitted light reaches a maximum at a certain temperature point and does not change. This is the temperature at which the TNBP is completely dissolved, and the molar fraction of solute at this temperature is the corresponding solubility.

### 3.3. Molecular Dynamics Simulation

The effect of intermolecular interactions on the solubility trends of TNBP in binary solvents was investigated using Materials Studio 8.0 software. The free energy of solvation was calculated to analyze the solute–solvent interactions [14,15]. The solute–solvent and solvent–solvent interactions were also analyzed by radial distribution functions (RDF) [16,17].

The TNBP crystal structures were taken from single crystal diffraction data, and the relevant solvent molecules (water, DMSO, acetone, acetonitrile, ethyl acetate) were plotted in the software. Firstly, the solute and solvent molecules were structurally optimized using the Forcite module under the COMPASS force field, and the structure was constructed using the Amorphous Cell module, including one TNBP molecule and 500 solvent molecules. Next, molecular dynamics simulations were carried out in the NVT system synthesis (the temperature of the system was controlled at 313 K using the NHL thermostat method, the integration step was set to 1 fs, and the total simulation time was 500 ps) [18,19]. The resulting equilibrium structural configuration can be directly analyzed to obtain the RDF curve in the state. Finally, the equilibrium structures are calculated separately for ideal, van der Waals and electrostatic free energies using the Forcite module and summed to obtain the total solvation free energy (100 ps for each contributing term).

## 4. Theoretical Foundations

### 4.1. Van’t Hoff Equation

The van’t Hoff equation [20,21,22] is constructed on the basis of the thermodynamic concept of solid–liquid equilibrium under the presumption that the solvent is an ideal liquid (γ = 1). The link between the inverse of the absolute temperature and the positive logarithm of the molar fraction solubility is expressed by this solubility equation. It accurately predicts the solute dissolved in the solvent over a finite temperature range. The equation is as follows:(5)$$\mathrm{lnx}=A+\frac{B}{T}\;$$
where T is the thermodynamic temperature, K; and A and B are model parameters.

### 4.2. Modified Apelblat Equation

The modified Apelblat equation, a semi-empirical model, is developed from the Clausius–Clapeyron equation. The equation provides a good representation of the relationship between temperature and solubility data in solvent systems [23,24,25,26].
(6)$$\mathrm{lnx}=A+\frac{B}{T}+\mathrm{ClnT}\;$$
where T is the thermodynamic temperature, K; and A, B and C are model parameters.

### 4.3. λh Equation

The λh model was first proposed by Buchowaski and is a common model for fitting experimental data using the λ and h parameters [27,28,29,30]:(7)$$\ln\left[1+\frac{\lambda \left(1\;-\;x\right)}{x}\right]=\lambda h\left(\frac{1}{T}-\frac{1}{T_{m}}\right)$$
where T is the thermodynamic temperature, K; and T_m_ is the melting point of TNBP, K; λ and h are the two parameters of the model.

### 4.4. Thermodynamic Model Evaluation

The mean relative deviation (ARD) and root mean square deviation (RMSD), which represent the fitting level of the thermodynamic model utilized, can be used to assess the applicability and accuracy of TNBP solubility data in pure solvent and mixed solvent systems [31].
(8)$$\mathrm{ARD}=\frac{1}{N}\overset{N}{\underset{i=1}{\sum }}\left|\frac{x^{\exp}-{\;x}^{\mathrm{cal}}}{x^{\exp}}\right|$$
(9)$$\mathrm{RMSD}=\sqrt{\frac{\sum_{i=1}^{N}{{(x}^{\exp}-{\;x}^{\mathrm{cal}})}^{2}}{N}}\;$$
where N is the number of experimental points; and x^exp^ and x^cal^ are the molar fraction solubility of TNBP’s experimental and calculated values, respectively.

### 4.5. KAT-LSER Model

The linear solvation energy equation (KAT-LSER), proposed by Kamlet and Taft et al., links solubility data with solvent characteristic parameters to investigate the effect of intermolecular interactions on solvation behavior. The expression is given by Equation (10) [32,33,34,35].
(10)$${\mathrm{lnx}=c}_{0}{+c}_{1}{\alpha +c}_{2}{\beta +c}_{3}\pi^{*\;}{+c}_{4}\left(\frac{V_{s}\delta_{H}^{2}}{100\mathrm{RT}}\right)$$
where α, β, π^*^, and δ_H_ stand for the hydrogen bond acidity, hydrogen bond alkalinity, dipole polarization rate, and Hildebrand parameter of the solvent, respectively. V_s_ represents the molar volume of the solute and can be estimated using the solute’s molar mass (M_1_) and density (ρ). c_0_ is the intercept of Equation (10). The values of c_1_ and c_2_ reflect changes in the nature of the solute’s interaction with the solvent through specificity hydrogen bonding. The values of c_3_ and c_4_ represent the solute’s sensitivity to non-specific electrostatic interactions between the solute and the solvent.

### 4.6. Thermodynamic Parameters

For the solid–liquid equilibrium system to be understood, it is required to compute the enthalpy of dissolution (Δ_dis_H), the entropy of dissolution (Δ_dis_S), and the Gibbs free energy (Δ_dis_G) using the van’t Hoff equation [36,37]. The equations are calculated as follows:(11)$$\mathrm{lnx}=-\frac{\Delta_{\mathrm{dis}}H}{\mathrm{RT}}+\frac{\Delta_{\mathrm{dis}}G}{R}$$

Assuming lnx is the dependent variable and 1/T is the independent variable, Δ_dis_H and Δ_dis_G can be calculated from the slope and intercept of the line fitted by the van’t Hoff equation, respectively. The equations are as follows:(12)$$\Delta_{\mathrm{dis}}H=-R\;\times \;\mathrm{slope}\;$$
(13)$$\Delta_{\mathrm{dis}}S=R\;\times \;\mathrm{intercept}$$

The following equation can be used to compute Δ_dis_G in various solvents:(14)$$\Delta_{\mathrm{dis}}G=\Delta_{\mathrm{dis}}H\;-{\;T}_{\mathrm{mean}}\Delta_{\mathrm{dis}}S$$
where the mean temperature is defined as follows:(15)$$T_{\mathrm{mean}}=\frac{N}{\sum_{N}^{i=1}\frac{1}{T_{i}}}\;$$
where T_i_ is the experimental temperature, K; and N is the total number of test points in each solvent.

## 5. Conclusions

The solubility of TNBP was investigated in six pure solvents and four mixed solvents using the laser monitoring technique at 0.1 MPa. In all solvent systems, the solubility of TNBP increased monotonically as the temperature rose. Moreover, the solubility in three solvent mixtures of DMSO with water, acetone, and acetonitrile increased with increasing DMSO content. The solubility in the DMSO + ethyl acetate mixture increased and then dropped with increasing mass fraction of DMSO, reaching the maximum at DMSO = 0.90 while remaining constant in temperature.

According to the KAT-LSER model, the hydrogen bonding alkalinity and self-cohesiveness of the pure solvent had a strong influence on the solubility of TNBP. The solubility of TNBP in mixed systems was analyzed at the molecular level using the solvation free energy and radial distribution function, concluding that the co-solvent phenomenon was dominated by solute–solvent interactions.

The three empirical equations used, including the van’t Hoff equation, modified Apelblat equation, and λh equation, all correlated well with the solubility data of TNBP. The modified Apelbla equation had the best applicability, except for the DMSO + water mixed system. Furthermore, the dissolution of TNBP in all solvent systems was a “heat-absorbing and non-spontaneous” process.

## Figures and Tables

**Figure 1 molecules-28-02424-f001:**
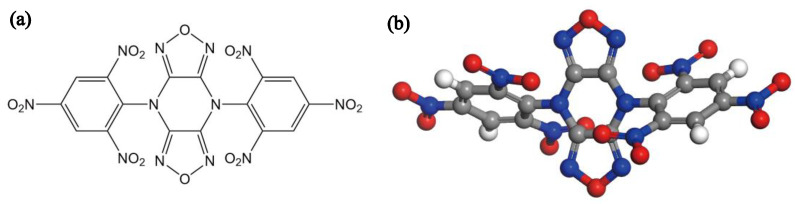
(**a**) The molecular structure of TNBP. (**b**) The 3D structure of TNBP.

**Figure 2 molecules-28-02424-f002:**
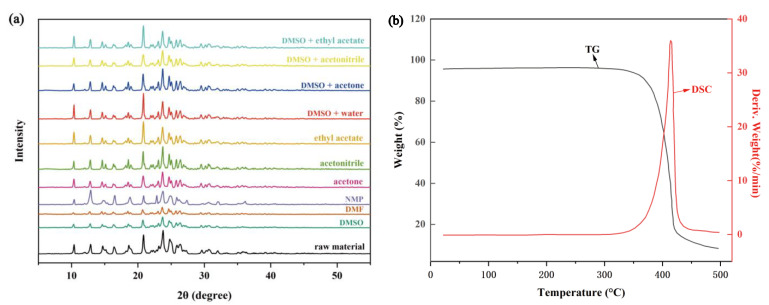
(**a**) PXRD patterns of TNBP raw material and crystalline samples obtained in different solvent systems. (**b**) TG-DTG curves of TNBP.

**Figure 3 molecules-28-02424-f003:**
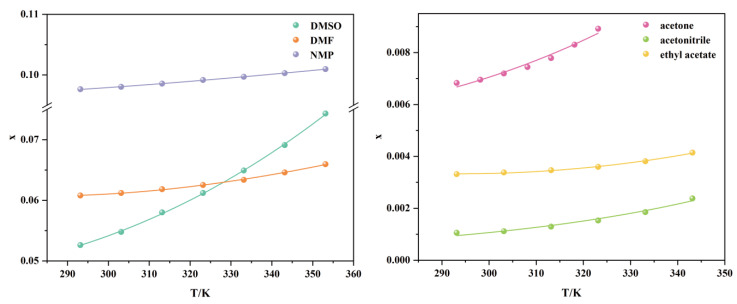
Modified Apelblat equation correlation of TNBP solubility data in six pure solvents.

**Figure 4 molecules-28-02424-f004:**
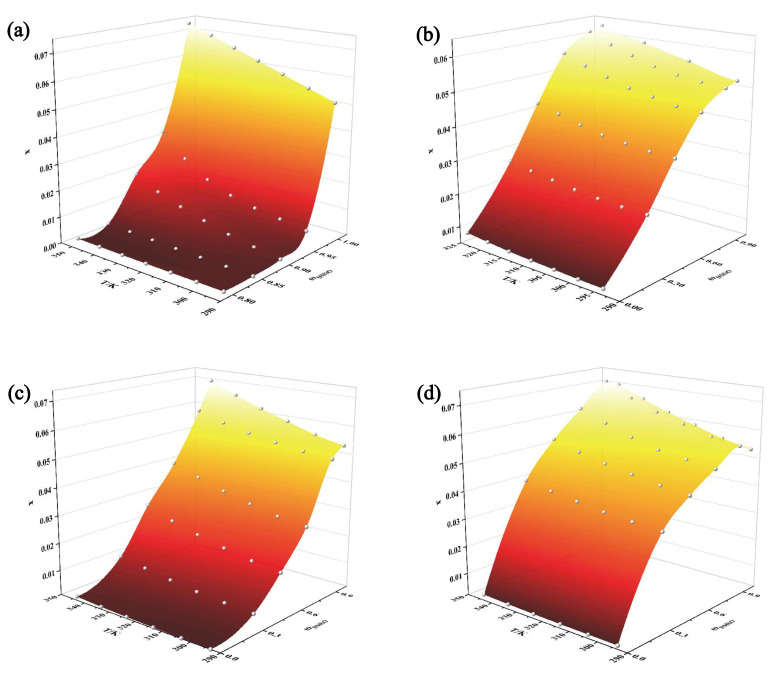
Experimental data on the solubility of TNBP in mixed solvents: (**a**) DMSO + water; (**b**) DMSO + acetone; (**c**) DMSO + acetonitrile; (**d**) DMSO + ethyl acetate.

**Figure 5 molecules-28-02424-f005:**
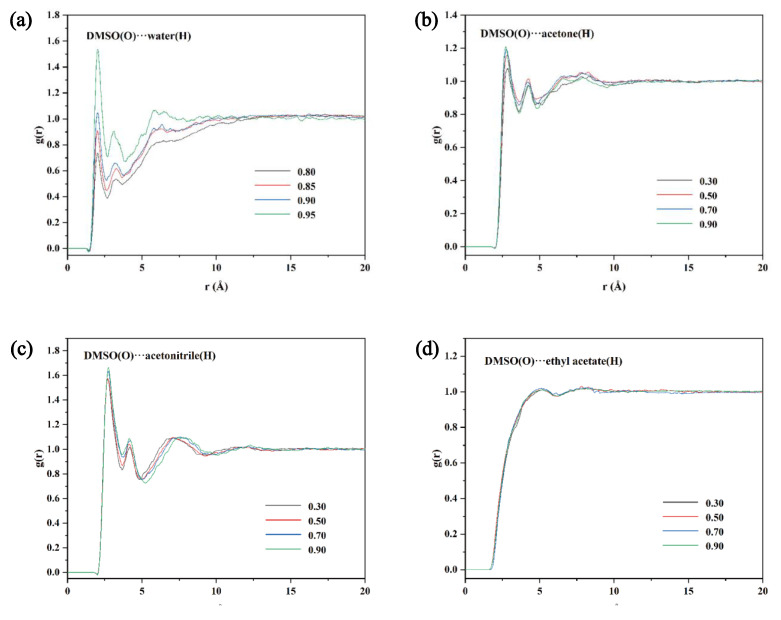
RDF plots of water (**a**), acetone (**b**), acetonitrile (**c**), and ethyl acetate (**d**) with different mass fraction of DMSO at 303.1 K.

**Figure 6 molecules-28-02424-f006:**
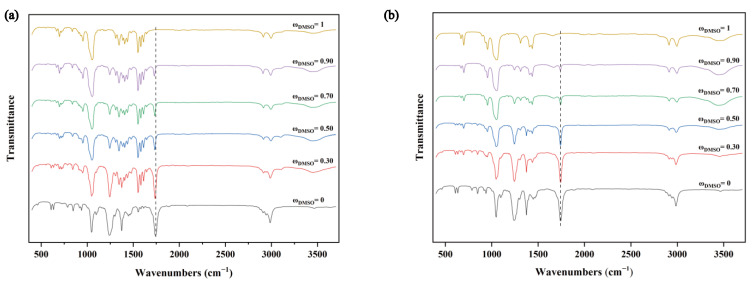
(**a**) The infrared spectrum of TNBP dissolved in DMSO + ethyl acetate mixture. (**b**) The infrared spectrum of DMSO + ethyl acetate solvent mixture.

**Figure 7 molecules-28-02424-f007:**
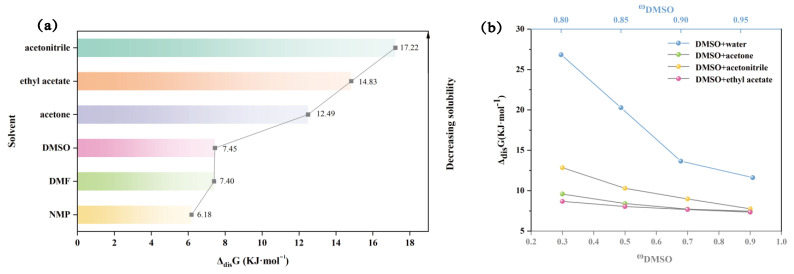
Δ_dis_G values for TNBP in pure (**a**) and mixed (**b**) solvents.

**Table 1 molecules-28-02424-t001:** Experimental molar fraction solubility and calculated solubility of TNBP in six pure solvents at 0.1 Mpa.

T	100×^exp^	100×^cal^		
		van’t Hoff	Modified Apelblat	λh
DMSO				
293.15	5.26	5.13	5.26	5.18
303.15	5.48	5.49	5.50	5.49
313.15	5.80	5.85	5.78	5.83
323.15	6.12	6.22	6.11	6.18
333.15	6.49	6.58	6.50	6.55
343.15	6.91	6.94	6.93	6.93
353.15	7.43	7.30	7.42	7.34
DMF				
293.15	6.08	6.03	6.09	6.06
303.15	6.12	6.12	6.12	6.12
313.15	6.18	6.21	6.18	6.19
323.15	6.25	6.30	6.25	6.27
333.15	6.34	6.38	6.35	6.36
343.15	6.46	6.46	6.46	6.46
353.15	6.60	6.53	6.59	6.57
NMP				
293.15	9.76	9.74	9.76	9.77
303.15	9.80	9.81	9.81	9.81
313.15	9.86	9.87	9.86	9.85
323.15	9.92	9.93	9.91	9.90
333.15	9.97	9.98	9.97	9.96
343.15	10.03	10.03	10.03	10.03
353.15	10.10	10.08	10.10	10.11
acetone				
293.15	0.683	0.661	0.667	0.662
298.15	0.695	0.694	0.694	0.694
303.15	0.720	0.728	0.724	0.727
308.15	0.744	0.762	0.757	0.761
313.15	0.779	0.797	0.793	0.796
318.15	0.830	0.832	0.832	0.832
323.15	0.892	0.868	0.875	0.869
acetonitrile				
293.15	0.106	0.091	0.095	0.092
303.15	0.112	0.112	0.113	0.112
313.15	0.129	0.136	0.134	0.136
323.15	0.153	0.163	0.160	0.162
333.15	0.185	0.193	0.191	0.193
343.15	0.238	0.226	0.229	0.227
ethyl acetate				
293.15	0.332	0.322	0.333	0.324
303.15	0.338	0.338	0.336	0.338
313.15	0.347	0.354	0.345	0.353
323.15	0.360	0.370	0.361	0.369
333.15	0.381	0.386	0.384	0.386
343.15	0.414	0.401	0.413	0.404

**Table 2 molecules-28-02424-t002:** Experimental molar fraction solubility and calculated solubility of TNBP in mixed solvents (DMSO + water) from 293.15 to 353.15 K at 0.1 Mpa.

T	100×^exp^	100×^cal^		
		van’t Hoff	Modified Apelblat	λh
ω_DMSO_ = 0.80				
293.15	0.000267	0.000549	0.000698	0.000549
303.15	0.00107	0.00119	0.00136	0.00119
313.15	0.00213	0.00244	0.00261	0.00244
323.15	0.00452	0.00480	0.00489	0.00480
333.15	0.00907	0.00907	0.00901	0.00907
343.15	0.0171	0.0165	0.0163	0.0165
353.15	0.0288	0.0290	0.0291	0.0290
ω_DMSO_ = 0.85				
293.15	0.00528	0.0144	0.0142	0.0144
303.15	0.0158	0.0231	0.0229	0.0230
313.15	0.0370	0.0358	0.0357	0.0358
323.15	0.0581	0.0540	0.0541	0.0540
333.15	0.0844	0.0795	0.0798	0.0795
343.15	0.116	0.115	0.115	0.114
353.15	0.158	0.162	0.161	0.162
ω_DMSO_ = 0.90				
293.15	0.109	0.183	0.196	0.182
303.15	0.250	0.285	0.293	0.284
313.15	0.431	0.430	0.432	0.430
323.15	0.675	0.634	0.629	0.634
333.15	0.949	0.912	0.901	0.913
343.15	1.29	1.29	1.28	1.29
353.15	1.76	1.78	1.79	1.78
ω_DMSO_ = 0.95				
293.15	0.699	0.516	0.575	0.516
303.15	0.862	0.727	0.761	0.727
313.15	1.01	1.00	1.01	1.00
323.15	1.25	1.35	1.33	1.36
333.15	1.64	1.80	1.76	1.80
343.15	2.25	2.35	2.32	2.35
353.15	3.17	3.02	3.06	3.01

**Table 3 molecules-28-02424-t003:** Experimental molar fraction solubility and calculated solubility of TNBP in mixed solvents (DMSO + acetone) from 293.15 K to 323.15 K at 0.1 Mpa.

T	100×^exp^	100×^cal^		
		van’t Hoff	Modified Apelblat	λh
ω_DMSO_ = 0.30				
293.15	2.18	2.12	2.14	2.12
298.15	2.19	2.20	2.20	2.20
303.15	2.24	2.27	2.26	2.27
308.15	2.31	2.35	2.34	2.35
313.15	2.41	2.43	2.42	2.43
318.15	2.50	2.51	2.51	2.51
323.15	2.64	2.59	2.61	2.59
ω_DMSO_ = 0.50				
293.15	3.46	3.37	3.40	3.38
298.15	3.49	3.49	3.49	3.49
303.15	3.56	3.62	3.60	3.61
308.15	3.66	3.74	3.72	3.73
313.15	3.81	3.86	3.85	3.86
318.15	4.00	3.99	3.99	3.99
323.15	4.19	4.11	4.14	4.12
ω_DMSO_ = 0.70				
293.15	4.52	4.40	4.44	4.41
298.15	4.56	4.58	4.58	4.58
303.15	4.70	4.75	4.73	4.75
308.15	4.84	4.93	4.90	4.92
313.15	5.04	5.11	5.08	5.10
318.15	5.28	5.28	5.28	5.28
323.15	5.56	5.46	5.50	5.47
ω_DMSO_ = 0.90				
293.15	5.01	4.87	4.92	4.88
298.15	5.06	5.05	5.05	5.05
303.15	5.15	5.23	5.21	5.23
308.15	5.29	5.41	5.38	5.40
313.15	5.53	5.59	5.57	5.59
318.15	5.76	5.78	5.78	5.78
323.15	6.09	5.96	6.00	5.97

**Table 4 molecules-28-02424-t004:** Experimental molar fraction solubility and calculated solubility of TNBP in mixed solvents (DMSO + acetonitrile) from 293.15 to 343.15 K at 0.1 Mpa.

T	100×^exp^	100×^cal^		
		van’t Hoff	Modified Apelblat	λh
ω_DMSO_ = 0.30				
293.15	0.57	0.52	0.56	0.53
303.15	0.61	0.62	0.62	0.62
313.15	0.69	0.72	0.70	0.72
323.15	0.81	0.84	0.81	0.84
333.15	0.96	0.96	0.95	0.96
343.15	1.12	1.10	1.13	1.10
ω_DMSO_ = 0.50				
293.15	1.57	1.49	1.57	1.50
303.15	1.69	1.70	1.70	1.70
313.15	1.87	1.92	1.87	1.91
323.15	2.10	2.15	2.09	2.14
333.15	2.36	2.40	2.37	2.39
343.15	2.72	2.65	2.72	2.66
ω_DMSO_ = 0.70				
293.15	2.80	2.71	2.79	2.73
303.15	2.93	2.96	2.95	2.96
313.15	3.15	3.21	3.15	3.19
323.15	3.40	3.46	3.39	3.44
333.15	3.70	3.72	3.69	3.71
343.15	4.04	3.97	4.05	3.99
ω_DMSO_ = 0.90				
293.15	4.91	4.82	4.91	4.84
303.15	5.00	5.01	4.99	5.01
313.15	5.13	5.20	5.13	5.18
323.15	5.31	5.39	5.31	5.37
333.15	5.54	5.57	5.55	5.57
343.15	5.85	5.75	5.84	5.78

**Table 5 molecules-28-02424-t005:** Experimental molar fraction solubility and calculated solubility of TNBP in mixed solvents (DMSO + ethyl acetate) from 293.15 to 343.15 K at 0.1 Mpa.

T	100×^exp^	100×^cal^		
		van’t Hoff	Modified Apelblat	λh
ω_DMSO_ = 0.30				
293.15	3.48	3.43	3.48	3.45
303.15	3.55	3.56	3.55	3.55
313.15	3.64	3.68	3.64	3.67
323.15	3.77	3.81	3.77	3.79
333.15	3.91	3.93	3.92	3.92
343.15	4.10	4.04	4.09	4.06
ω_DMSO_ = 0.50				
293.15	4.35	4.28	4.35	4.31
303.15	4.47	4.49	4.47	4.48
313.15	4.65	4.69	4.63	4.67
323.15	4.82	4.89	4.83	4.87
333.15	5.06	5.08	5.07	5.07
343.15	5.35	5.27	5.34	5.30
ω_DMSO_ = 0.70				
293.15	4.97	4.83	4.97	4.86
303.15	5.09	5.10	5.08	5.10
313.15	5.25	5.37	5.26	5.35
323.15	5.52	5.63	5.53	5.61
333.15	5.88	5.89	5.86	5.89
343.15	6.27	6.15	6.28	6.18
ω_DMSO_ = 0.90				
293.15	5.50	5.36	5.50	5.39
303.15	5.65	5.69	5.67	5.69
313.15	5.94	6.02	5.91	5.99
323.15	6.23	6.34	6.23	6.32
333.15	6.61	6.67	6.63	6.66
343.15	7.13	6.98	7.12	7.02

**Table 6 molecules-28-02424-t006:** The solvation free energy of TNBP in the four solvent mixtures at 313.15 K.

ω_DMSO_	ΔG_solv_ (Kcal·mol^−1^)	ω_DMSO_	ΔG_solv_ (Kcal·mol^−1^)		
	DMSO + Water		DMSO + Acetone	DMSO + Acetonitrile	DMSO + Ethyl Acetate
0.80	−7.30	0.30	−10.69	−8.91	−11.41
0.85	−7.77	0.50	−12.33	−10.29	−13.85
0.90	−8.51	0.70	−16.94	−14.05	−19.15
0.95	−9.89	0.90	−21.52	−20.67	−23.47
1.00	−22.77	1.00	−22.77	−22.77	−22.77

## Data Availability

The data presented in this study are available to all readers according to “MDPI Research Data Policies”.

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
