# Peer review of "Solubility Determination and Comprehensive Analysis of the New Heat-Resistant Energetic Material TNBP"

_molecules, 2023, doi:10.3390/molecules28062424_

Round 1

Reviewer 1 Report

The authors described the solubility analysis for TNBP, which is important for its application. The solubility is analyzed by laser monitoring, more detail should be provide for this technique, and what is advantage is this method compared to the traditional methods?

Reviewer 2 Report

The paper «Solubility Determination and Comprehensive Analysis of the New Heat-resistant Explosive TNBP» by Minchang Wang et al. is related to the experimental and theoretical studies of solubility of TNBP in several pure solvents and various mixtures of DMSO with some polar solvents. The experimental studies are scientifically meaningful and they are supported by molecular dynamics calculations. The experimental data on solubility were fitted with three equations and semi-empirical Apelbladt model has demonstrated the best applicability. The paper leaves generally positive impression, however, there are some issues that should be addressed to authors:

1. Figure 4. In caption to this figure one can see that plots are related to data on the solubility of FOX-7. At the same time in text (line 113) is alleged that plots on Figure 4 are related to TNBP. Authors should correct either text or Figure caption.

2. According to Tables 2-5 the modified Apelblat equation fits experimental data somewhat better than other two equations. Unfortunately, it is hard to analyze the quality of fit with only Tables. Authors should give additional figures with experimental data and fitted equations. Besides, the quantitative information like R2 should be supplied with these plots.

3. In addition of RDF it will be more fruitful to analyze the hydrogen bonding with vibration spectra that can be routinely calculated form velocity auto-correlation function form MD simulations. In turn, calculated vibrational spectra can be compared with IR spectra of solutions to additionally justify the conclusions of the present paper.

To accept this paper for publication authors should consider above issues and make some additional work to make their conclusion more solid.

Reviewer 3 Report

The work is performed competently into crystal and solubility of an explosive, 4,8-Di(2,4,6-trinitrophenyl) difurazano[3,4-b:3’,4’-e]pyrazine (TNBP); the results are plausible, and I appreciate that the authors share their results freely and openly, which benefits the military organizations of all countries that obviously have interest in explosives. But I also feel uncomfortable to have to read through a manuscript clearly aiming at military application which will lead to the increase in deaths of people, eventually.

Round 2

Reviewer 2 Report

I satisfied the replies, so now the article can be accepted for publication in Molecules.